# Beyond the Visible: A Review of Ultraviolet Surface-Enhanced Raman Scattering Substrate Compositions, Morphologies, and Performance

**DOI:** 10.3390/nano13152177

**Published:** 2023-07-26

**Authors:** Andrea N. Giordano, Rahul Rao

**Affiliations:** 1Materials and Manufacturing Directorate, Air Force Research Laboratory, Wright-Patterson Air Force Base, Dayton, OH 45433, USA; 2National Research Council, Washington, DC 20001, USA

**Keywords:** ultraviolet surface-enhanced Raman scattering (UV-SERS), ultraviolet surface-enhanced resonant Raman scattering (UV-SERRS), deep-ultraviolet surface-enhanced Raman scattering (DUV-SERS), deep-ultraviolet surface-enhanced resonant Raman scattering (DUV-SERRS), UV-SERS substrates, enhancement factor (EF), limit of detection (LOD), photoluminescence (PL), sensors

## Abstract

The first observation of ultraviolet surface-enhanced Raman scattering (UV-SERS) was 20 years ago, yet the field has seen a slower development pace than its visible and near-infrared counterparts. UV excitation for SERS offers many potential advantages. These advantages include increased scattering intensity, higher spatial resolution, resonance Raman enhancement from organic, biological, and semiconductor analytes, probing UV photoluminescence, and mitigating visible photoluminescence from analytes or substrates. One of the main challenges is the lack of readily accessible, effective, and reproducible UV-SERS substrates, with few commercial sources available. In this review, we evaluate the reported UV-SERS substrates in terms of their elemental composition, substrate morphology, and performance. We assess the best-performing substrates with regard to their enhancement factors and limits of detection in both the ultraviolet and deep ultraviolet regions. Even though aluminum nanostructures were the most reported and best-performing substrates, we also highlighted some unique UV-SERS composition and morphology substrate combinations. We address the challenges and potential opportunities in the field of UV-SERS, especially in relation to the development of commercially available, cost-effective substrates. Lastly, we discuss potential application areas for UV-SERS, including cost-effective detection of environmentally and militarily relevant analytes, in situ and operando experimentation, defect engineering, development of materials for extreme environments, and biosensing.

## 1. Twenty Years of Ultraviolet Surface-Enhanced Raman Scattering

Since its discovery 49 years ago, surface-enhanced Raman scattering (SERS) has developed a diverse research portfolio in the fields of the physical sciences, life sciences, materials science, and engineering, just to name a few. The ultrasensitive detection enabled through the SERS mechanisms (see Section 2) can be utilized to develop analytical techniques for identifying and quantifying analytes. These techniques include chemosensors, enantioselective discrimination of chiral molecules, chemometrics for quantification, chemobiosensors, molecular beacons, and the detection of DNA, biomarkers, tumor cells, and pathogens [1]. Additionally, SERS has provided kinetic and mechanistic insights into heterogeneous catalytic systems using in situ and operando experiments [2]. To gain a fundamental understanding of structure–property relationships in novel materials, SERS has been used for various purposes. These include surface characterization, molecular composition, and surface bonding of interaction species of materials. Additionally, SERS has been employed to investigate the growth kinetics of films, mechanically induced conformational changes, ion-intercalation processes, corrosion, and electrocatalysis [3]. At the time of this writing, SERS has been the subject of approximately 799,000 journal articles, review articles, books, book chapters, theses, dissertations, retractions, conference proceedings, patents, and editorial perspectives, as evidenced by a Google Scholar search. However, most of these studies focused on visible and near-infrared (NIR) excitation. When narrowing the search to ultraviolet (UV) excitation, a Google Scholar search returns only 285 results. The field of UV-SERS has seen slower growth compared to its visible and NIR counterparts, with only a handful of publications each year since it was first reported 20 years ago (Figure 1) [4]. To the best of our knowledge, there have been a limited number of book chapters and review articles dedicated to UV-SERS [5,6,7,8]. Only a handful of SERS review articles discuss UV-SERS [3,9,10,11,12,13,14], and no reviews of UV-SERS substrates provide the motivation for this review.

Van Duyne and coworkers attributed the slower pace of UV-SERS advancement to three main factors [2]:Incompatibility of existing commercially available visible–NIR instrumentation with UV optics, requiring a separate, typically home-built system that reduces accessibility;Poor plasmonic properties of typical SERS substrate metals like Ag and Au, thus requiring exploration and fabrication of UV-SERS substrates;Photodegradation of analytes with UV excitation.

As instrumentation technology advances (optics, detectors, laser designs, etc.), UV-SERS instrumentation is becoming accessible to more researchers, reducing this obstacle. Fabrication of effective UV-SERS substrates remains a challenge, and this review will focus on the performance of reported UV-SERS substrates. The last challenge, sample damage from UV excitation, will remain a disadvantage of UV-SERS over its visible and NIR analogs.

While these limitations may have slowed the pace of UV-SERS development, there is motivation to explore the potential capabilities that UV excitation sources provide. From a capability perspective, the UV excitation source equips researchers with the ability to access Raman scattering enhancement through the overlap of the excitation source with an UV electronic transition of an analyte (resonance Raman, discussed in more detail in Section 2). Since many aromatic, biological, and semiconductor materials absorb strongly in the UV region, UV-SERS is advantageous for low-concentration detection of these analytes. Additionally, metal-enhanced fluorescence (MEF) can occur when there is an overlap of the analyte’s UV absorption with the substrate’s localized surface plasmon resonance (LSPR) [15]. Simultaneous MEF and SERS can be collected when a large enough spectral window is used, but many researchers choose to collect them separately. In addition to enhancing Raman scattering and fluorescence, the UV excitation source can initiate photochemical and photocatalysis reactions for in situ (or potentially operando) monitoring of the reaction kinetics and mechanism. The SERS enhancement will play a vital role in in situ monitoring of low-concentrations of reactants or products. One of the main disadvantages of UV excitation sources is the potential for damage to your analyte, but this can be an advantage in degradation or defect engineering studies, especially in materials science.

In this review, we will explore the current progress on developing UV-SERS substrates. This includes the variety of substrate compositions, morphologies, and performances for both experimentally reported and theoretically predicted substrates. In Section 2, we will provide a brief overview of Raman spectroscopy and enhancement techniques, including comparing the advantages and disadvantages of UV-SERS in comparison with visible–NIR SERS. Section 3 includes an analysis of the reported substrate composition and morphology combinations and the best-performing substrates in terms of their enhancement factors (EFs) and limits of detection (LODs). Additionally, we discuss the challenges and opportunities for UV-SERS substrates. The last section discusses future directions and potential applications for UV-SERS.

## 2. Overview of Raman Spectroscopy and Enhancement Techniques

Raman spectroscopy is a well-developed technique, with over two million publications cited on Google Scholar since the discovery of Raman scattering in 1928 by Sir Chandrasekhara Venkata Raman [16]. Raman spectroscopy detects the inelastic scattering (Stokes and anti-Stokes) of the incident radiation resulting from the vibrational modes in molecules and materials. The vibrational modes have characteristic frequencies from the chemical bonds and phonons in molecules and materials, respectively. These characteristic frequencies lead to a vibrational fingerprint signature sensitive to external factors, including temperature, pressure, and molecular interactions (i.e., solvation, isotopic substitution, material defects, material strain, and protein binding). This sensitivity has led to the widespread use of Raman spectroscopy for identification and characterization across the fields of science and engineering [1,6,17]. We point the reader to several excellent books for a more detailed description of Raman spectroscopy [18,19,20,21,22,23,24].

A major challenge with Raman spectroscopy is the low probability of inelastically scattered photons, with approximately 1 in 10^8^ incident photons being inelastically scattered [17]. The intensity of the Raman signal is proportional to the Raman cross-section (approximately 10^−30^ cm^2^ sr^−1^ molecule^−1^) and the power density of the excitation laser [2,25]. One way to overcome this obstacle is to enhance the Raman intensity through a variety of enhancement techniques, including resonance Raman scattering (RRS), SERS, and surface-enhanced resonance Raman scattering (SERRS). There are many other enhancement techniques, such as tip-enhanced Raman scattering (TERS), which are beyond the scope of this review. RRS requires a resonance between the analyte’s electronic transition and the excitation laser energy. The excitation of the electronic transition results in a larger Raman cross-section that produces intensities from 10^2^ to 10^6^ times larger than nonresonant intensities [25,26,27]. SERS requires the excitation laser energy to be resonant with the local surface plasmon of a metal substrate. The SERS mechanisms for intensity enhancement are derived from electromagnetic (EM) field enhancement and chemical enhancement (charge transfer), as described in many reviews and references therein [1,2,3,14,28]. Fleischmann et al. reported the first observation of SERS in 1974, detecting an increase in the intensity of pyridine adsorbed on a roughened silver electrode [29]. In the last 49 years, there have been almost half a million publications on the topic, according to Google Scholar. Typical SERS EFs range from 10^4^ to 10^8^, with enhancements from the EM mechanism ranging from 10^4^–10^7^ and from the chemical mechanism ranging from 10–100 [2,30]. Additional enhancement can occur when the local surface plasmon of the SERS substrate and the electronic transition of the analyte are resonant with the excitation laser energy through SERRS. In 1977, the first report of SERRS investigated the enhancement of pyridine intensities under resonance conditions on a roughened silver electrode [31]. Both SERS and SERRS have allowed for single-molecule detection of crystal violet (nonresonant) and rhodamine 6G (resonant), with EFs reported up to 10^14^ [32].

There are many common excitation lasers that span energies from the deep ultraviolet (DUV) to the NIR, allowing RRS, SERS, and SERRS to be utilized by a wide variety of substrates and analytes. There are advantages and disadvantages to SERS with every laser excitation energy, so Raman spectroscopists want as many laser lines as possible. While the majority of SERS studies have focused on visible and NIR excitations, the DUV–UV region (190–380 nm) remains a less explored field. Here, we review the advantages and disadvantages of SERS in the DUV–UV region and critically assess the scientific literature involving high-energy UV laser excitation for SERS. For ease of reading, we will use UV to refer to the entire DUV–UV region but will specify excitation wavelengths as appropriate when discussing SERS substrate performance.

### 2.1. Advantages and Disadvantages of UV-SERS

There are several advantages to UV excitation over visible excitation. These include increased Raman scattering efficiencies, higher spatial resolution, potential resonance overlaps for analytes, probing UV photoluminescence (PL), and mitigating visible PL from analytes and substrates due to strong absorption in the UV. The Raman scattering intensity varies with the fourth power of the laser excitation frequency. This leads to higher scattering efficiencies as the laser excitation frequency increases towards the UV region [2]. The direct dependence of laser excitation wavelength on diffraction-limited spatial resolution also allows for higher spatial resolution in the UV [2]. Many organic and biological analytes absorb strongly in the UV region, leading to the potential for SERRS detection of these analytes with the appropriate UV-SERS substrate. Furthermore, the UV absorption of the analyte allows for the ability to measure PL during the experiment. Lastly, the UV excitation is high enough in energy to separate the PL from the Raman signal, which mitigates the problem of the PL signal interfering within the Raman spectrum, such as the PL of rhodamine 6G or a glass slide at 532 nm and 785 nm excitations, respectively. Many of these advantages are analyte-specific, thus requiring consideration on a case-by-case basis.

While the advantages are analyte-specific, the disadvantages of UV-SERS stem from the UV lasers and the substrates. Many of the commercially available UV lasers fall into high- or low-power classes. The high-power lasers (>1 mW) produce more signal but are dangerous and expensive compared to the low-power lasers (<1 mW), which produce less signal but are safer and cheaper. While powerful lasers of all wavelengths are dangerous, UV lasers are particularly damaging to the cornea and lens of the eye. In contrast, visible/NIR lasers are more damaging to the retina [33]. Additionally, the high energy of the UV lasers can potentially cause photodegradation damage to the analyte, especially in biological and polymeric materials. The largest disadvantage is the lack of well-characterized and reproducible substrates in the literature and the limited availability of high-quality commercial substrates. However, this disadvantage also provides the greatest opportunity for improvement as the field of UV-SERS matures. In the following section, we will focus on SERS substrate performance in the UV region.

## 3. UV-SERS Substrate Performance

When selecting an optimal UV-SERS substrate material for an analyte, one must match its LSPR with the available UV excitation source(s). The elemental composition and the morphology (size and shape) affect the LSPR of the substrate, as shown in Figure 2. Considering the most common visible SERS substrates composed of Au, Figure 2 illustrates the LSPR ranges from 400 nm into the NIR. This signifies that the elemental composition of the substrate affects the range of the LSPR. From there, the substrate morphology can narrow this range. For example, spherical Au nanoparticles (NPs) can have an LSPR range of 500–650 nm. The size of the spherical NP will tune the LSPR within that range, ideally with a maximum resonance with the visible lasers in that range (i.e., 514, 532, 633 nm, etc.). Using Figure 2 as a starting point, we can see that Al, Mg, and In have an LSPR within the UV region. In the next section, we will explore the variety of elemental compositions and morphologies used in UV-SERS substrates.

### 3.1. Elemental Compositions and Morphologies of UV-SERS Substrates

The first reported UV-SERS observation was on roughened Rh and Ru electrodes using λ_ex_ = 325 nm to detect pyridine and thiocyanate 2003 [4]. For the next seven years, reports continued to explore UV-SERS on other roughened electrodes, including Au, Co, Pt, and Pd, as shown in Figure 3 [35,36,37,38,39,40]. The first reports moving away from using electrodes for UV-SERS used Cu NPs with λ_ex_ = 325 nm to detect p-hydroxybenzoic acid [41] and an Al nanostructure (λ_ex_ = 244 nm) for SERRS detection of crystal violet in 2007 [42]. From 2008 to the present, researchers have been exploring different combinations of elemental composition and substrate morphology, as illustrated in Figure 3. The variety of elemental compositions explored encompasses the traditional coinage metals (Au, Ag, Cu) [36,37,39,40,41,43], transition metals (Co, Pd, Pt, Rh, Ru) [4,35,38,43,44,45], Group 3A metals (Al, Ga, In) [42,46,47,48,49,50,51,52,53,54,55,56,57,58,59,60,61,62,63,64,65,66], 2D materials (SnS_2_, SnSe_2_) [67,68], alloys (Al–Mg) [69,70], and oxides (ZnO) [71]. We defined five substrate morphology categories to simplify the analysis: 2D materials, electrodes, NPs, and nanostructures (NSs). We have included spherical, non-spherical, and core-shell particles for the NP category. For NSs, we have single-element structures, including nanohole arrays (NHAs), nanovoid arrays (NVAs), and nanocavity arrays (NCAs), in addition to multicomponent structures with a film over nanosphere (FON) or self-assembled NPs on top of the substrate. Figure 4 provides some examples of the variety of NSs evaluated. With the variety in elemental composition and substrate morphology, many possible combinations have shown UV-SERS activity, as visualized by the thickness of the colored links in Figure 3. The most studied UV-SERS substrate has been Al NSs, with 19 published reports [42,46,47,48,49,50,51,52,53,54,55,56,57,58,59,60,61,62,63] (highlighted with thick, blue link in Figure 3).

In the next section, we will explore the SERS performance metrics to determine the best-performing UV-SERS substrates. We intentionally omitted the UV-TERS substrates from our comparison, as the enhancement mechanism and instrumentation are more complex. However, here are a few reports on UV-TERS [5,72,73,74,75] for interested readers. Additionally, we only analyzed substrates with demonstrated UV-SERS performance and excluded substrates with only plasmonic properties in the UV but no reported SERS measurements.

### 3.2. UV-SERS Substrate Performance

The EF and the LOD are two complementary performance metrics for evaluating SERS substrates. The EF quantifies the signal enhancement of the analyte from the substrate as a ratio of the SERS intensity (I_SERS_) to the normal Raman intensity (I_NRS_), each normalized by the number of molecules in the measurement (N_SERS_ and N_NRS_), as shown in Equation (1) [2].
(1)$$\mathrm{EF}=\frac{I_{\mathrm{SERS}}/N_{\mathrm{SERS}}}{I_{\mathrm{NRS}}/N_{\mathrm{NRS}}}$$

Van Duyne and coworkers noted the challenges associated with accurately determining N_SERS_ and N_NRS_, including illuminated volume (spot size and depth of focus) and surface coverage of adsorbed molecules on a substrate, which makes direct comparisons of EFs non-trivial [2]. Another complicating factor for comparing EFs is the resonant enhancement from the analyte when measured under resonant conditions (i.e., comparing SERRS to SERS), especially in the UV, where many organic and biological analytes have absorptions in this region. The second performance metric often reported is the LOD, which quantifies the lowest concentration of the detectable analyte. As with EFs, there are challenges with standardizing LOD, as the measurement is highly dependent on the spectrometer conditions, such as laser power, exposure time, and number of accumulations. While there are known issues with reporting accurate EFs and LODs, they are the most common performance metrics reported for new SERS substrates. We will use these performance metrics to make general comparisons between the reported UV-SERS substrates, denoting the use of a resonant analyte. In Figure 5, we have compiled the top-performing UV-SERS substrates with reported EFs >10^4^ and LODs when available. The best-performing substrate would have the highest EF (largest blue bar) and the lowest LOD (smallest orange-striped bar). We will break the substrates into two groups: DUV excitation (λ_ex_ = 229–266 nm) and UV excitation (λ_ex_ = 325 nm). In the subsections below, we will discuss the top four substrates in the DUV (Section 3.2.1) and UV (Section 3.2.1) groups.

#### 3.2.1. DUV-SERS Substrate Performance

In considering the DUV-SERS substrates in Figure 5, the EFs range from 10^5^–10^7^, and the only LOD reported was 10^−13^ M under resonant conditions, most commonly using adenine as an analyte. We highlight four substrates, including an Al film over nanosphere (FON), an Al nanovoid array (NVA), an Al nanohole array (NHA), and Rh concave nanocubes (CNC). The best-performing substrate in the DUV region was the Al FON substrate, with the highest reported EF ~10^7^ at λ_ex_ = 229 nm [49]. This substrate utilized drop-casting 170 nm silica spheres or 210 nm carboxylated latex/polystyrene spheres onto a Si wafer, followed by vapor deposition of 200 nm of Al to create the Al FON substrate (Figure 4a). The SERS performance of the Al FON was compared against Al film on Si, Ag FON, and Ag film on Si using resonant (trans-1,2-bis(4-pyridyl)-ethylene, tris(bipyridine)ruthenium(II), and adenine) and nonresonant (6-mercapto-1-hexonal) analytes. The best-performing substrate was the 210 nm Al FON, with the highest SERRS EF of 2.76 × 10^7^ and the highest SERS EF of 2.00 × 10^5^ (Figure 6a). We decided to include the SERRS EF for this substrate in our analysis to allow for a better comparison since all the other studies used resonant analytes and only reported a SERRS EF.

Next, we highlight two Al NS arrays, an Al nanovoid array and an Al nanohole array, both exhibiting high EFs ~10^6^. Sigle et al. produced the Al nanovoid array substrate by depositing Al over a template of a self-assembled monolayer of polystyrene spheres of various sizes on PMMA sheets [62]. Then, the spheres were dissolved in THF to leave behind hemispherical voids with diameters ranging from 100–500 nm (Figure 4b). The SERS signal for 1 mM adenine for all five diameter voids was measured at resonant (244 nm) and nonresonant (785 nm, Klarite substrate) conditions (Figure 6b). Compared with a thin film of Al evaporated on a silicon wafer and an adenine solution without a plasmonic surface as a control, the 100 nm cavity produces the largest SERRS EF ~10^6^ due to the stronger localization of the field in a small volume located along the rim determined through finite difference time domain (FDTD) simulations. For the next substrate, Dubey et al. designed an Al nanohole array using FDTD analysis to optimize its LSPR for λ_ex_ = 266 nm [53]. The design optimization fixed the nanohole diameter at 100 nm and allowed the periodicity to range from 150–250 nm, with 200 nm optimal. To create the Al nanohole array, first, a 150 nm thick Al film was epitaxially grown on sapphire, and then the holes were created using electron beam lithography and a reactive ion etching process that produced 100 nm diameter holes with a 200 nm periodicity (Figure 4c). The Al nanohole array substrate performance was assessed by sublimating a uniform and ultrathin (~1 nm) film of five nucleotide monomers (adenine, thymine, cytosine, guanine, and uracil) and detecting SERRS at 266 nm with EF up to 10^6^ (Figure 6c). Additionally, the nanohole arrays could detect and distinguish between mutations in oligonucleotides, such as 12-mer single-stranded DNA.

Lastly, we highlight Rh NPs for their reasonable EF of ~10^5^ and excellent LOD of ~10^−13^ M [45]. Thus far, this is the only non-Al substrate used in our analysis for DUV-SERS studies. A variety of Rh NPs were synthesized, including triangular nanoplates (TNPs, d = 7 nm), rectangular nanoplates (RNPs, AR 1.4), and concave nanocubes (CNCs, l = 27 nm). To evaluate their SERRS performance at λ_ex_ = 266 nm, the Rh NPs were drop-casted onto a silicon substrate, followed by drop-casting 1 µM adenine solution (Figure 6d). The Rh CNCs performed the best with an EF of 4.5 × 10^5^, but the Rh TNPs and Rh RNPs had comparable performance with EFs of 2.2 × 10^5^ and 2.5 × 10^5^, respectively. Moreover, the SERS performance of the Rh NPs at nonresonant conditions was evaluated at λ_ex_ = 785 nm using 10^−5^ M rhodamine 6G, with EFs ranging from 2.1 × 10^3^–9.9 × 10^3^, with the Rh CNCs continuing to perform the best. Due to the highest EF of the Rh CNCs under DUV excitation, the LODs for trace levels of three explosive molecules, p-nitrobenzene, 2,4-dinitrotoluene, and ammonium nitrate, were determined to be 1.3 × 10^−10^ M, 1.1 × 10^−7^ M, and 4.9 × 10^−13^ M, respectively.

#### 3.2.2. UV-SERS Substrate Performance

In the UV region of Figure 5 (λ_ex_ = 325 nm), we see reported EFs ranging from 10^5^–10^7^ at nonresonant conditions and LODs ranging from 10^−6^–10^−16^ M. We highlight four substrates, including a self-assembled Al NS, an Al nanocavity array (NCA), an Al 3D hybrid NS, as well as a self-assembled NS of In NPs coated with SiO_2_. The best-performing substrate in the UV region was the self-assembled Al NSs [56]. This substrate was produced through Al deposition and annealing to form stalagmite-shaped Al NPs on a quartz substrate (Figure 4d). The effect of deposition thickness and annealing temperature was explored on the SERS activity of these substrates at λ_ex_ = 325 nm with 1 mM adenine under nonresonant conditions (Figure 7a). The highest reported EF was 3.49 × 10^7^ for an Al NS with a 6 nm deposition thickness without an annealing treatment. This was also the highest reported EF of all the substrates analyzed in the DUV and UV regions. The reported LOD for this substrate was 1 × 10^−7^ M.

Next, we would like to highlight two substrates, an Al 3D hybrid NS and an Al nanocavity array, both of which report the next highest EFs ~10^6^. Li et al. fabricated the Al 3D hybrid NSs from 3D-stacked hybrid assemblies of Al@Al_2_O_3_ core-shell NPs of diameters of 20 nm and 50 nm using millisecond laser direct writing in liquid nitrogen (Figure 4e) [46]. The SERS performance was tested using 50 µM crystal violet at λ_ex_ = 325 nm (nonresonant) and compared to Al NSs processed in air and water (as opposed to liquid nitrogen), bare Al plate, and sapphire as controls (Figure 7b). The 3D hybrid NS processed in liquid nitrogen outperformed the other substrates with a reported EF of 2.81 × 10^6^ and a LOD of 5 µM. The performance of the 3D hybrid (20 and 50 nm particles) NS was evaluated using FDTD simulations. This indicates hot spot generation between the large NPs decorated with the small NPs and between the small NPs with an ideal dielectric gap of 3.0 nm thick interlayer Al_2_O_3_ shell separating NPs. To create the Al nanocavity array substrate, Zeng et al. used self-assembled polystyrene spheres on a Si wafer, followed by a deposition of 50 nm of Al using an electric beam evaporation system [59]. Then a stripping process removed the polystyrene spheres, leaving an Al nanovoid array, before the final step of etching the cavities using inductively coupled plasma to leave an array of nanocavities with depths ranging from 100–400 nm (Figure 4f). For comparison, a variety of control substrates were created, including Al nanovoid arrays (no etching), Au nanocavity arrays, Au nanovoid arrays (no etching), Al film, and Au film. To measure the SERS performance, 1 mM adenine at λ_ex_ = 325 nm (nonresonant) was used to compare all the substrates (Figure 7c). The 300 nm Al nanocavity outperformed all the other substrates, with an EF of 1.3 × 10^6^. To explain the superior performance of the 300 nm Al nanocavity, the reflectance spectra showed a plasmonic resonance of 338 nm, close to the 325 nm excitation.

The last substrate we highlight is a 2D self-assembly of In@SiO_2_ NP NS for its superior LOD (10^−16^ M) with a reasonable EF ~10^5^ [65]. Das et al. synthesized the In@SiO_2_ core-shell NPs using the Stöber method to deposit silica shells with a controlled thickness onto the spherical In NP cores. The average In spherical NP diameter was 40 nm, and the silica shell thickness ranged from 2 nm to 20 nm. The NS was constructed by self-assembling the In@SiO_2_ NPs onto a glass substrate functionalized with (3-aminopropyl)trimethoxysilane (Figure 4g). The SERS performance for these substrates was measured with 1 mM of adenine and tryptophan at λ_ex_ = 325 nm (nonresonant) and compared with bare In and a glass reference (Figure 7d). The SERS performance of the silica shell was examined. The results determined that the SERS signal decreased with increasing shell thickness, leading to the 2 nm thick silica shell particles performing the best with EFs of 6.3 × 10^5^ for tryptophan and 4.3 × 10^5^ for adenine. The LOD for tryptophan was 0.18 fM for these particles, and that for adenine was 0.04 fM. The study also reported on the stability of these substrates for up to 4 months of exposure to the atmosphere compared to the bare In NPs, which showed rapid degradation when exposed to the atmosphere.

#### 3.2.3. Novel and Emerging Experimental Studies and Numerical Simulations

The subsections above highlighted some of the top-performing UV-SERS substrates. We would also like to highlight a few reports exploring novel substrate composition and morphology combinations, such as SnS_2_ and SnSe_2_ 2D materials, along with Al-Mg alloys and ZnO NSs. Although these substrates did not demonstrate exemplary SERS performance, they warrant mention for potential future exploration of UV-SERS substrates. Semiconductor 2D materials offer unique potential advantages for UV-SERS substrates through band gap engineering to match laser excitation and an enhanced charge-transfer process between the analyte and the 2D material [67]. Thus far, carbon-doped SnS_2_ nanoflowers [67] were reported to exhibit an EF of 2.9 × 10^2^ and a LOD of 10^−7^ M, while SnSe_2_ nanoflakes [68] only reported a LOD of 10^−7^ M using crystal violet with λ_ex_ = 325 nm. Next, the first reported UV-SERS alloy was an Al_x_–Mg_1−x_ film (x = 0.16), demonstrating a maximum EF of 9.5 using adenine at λ_ex_ = 257 nm (resonant conditions) [70]. Lastly, CVD-grown ZnO microrod NSs with sizes ranging from 220–633 nm were examined for SERS activity using 4-mercaptopyridine, where λ_ex_ = 214 nm showed the largest enhancement with an EF of 37.8 [71]. While these examples do not have the best UV-SERS performance, they illustrate the range of UV-SERS substrates being explored experimentally.

In addition to experimental studies, researchers are investigating different types of UV-SERS substrates using numerical simulations, such as FDTD or finite element methods. Only a handful of reports have explored substrate composition and morphology, mostly focusing on Al NP dimers (size, shape, and interparticle spacing) [76,77,78,79]. Additionally, there are a few reports on Al NSs [80,81] and one on Al complexes and junctions with pyridine [82]. Thus far, there have been four studies investigating the UV plasmonic behavior of metal NPs (Al, Cr, Cu, Ga, In, Mg, Pd, Pt, Rh, Ru, Ti, W) [83,84], homodimers of poor metals (Al, Ga, In, Sn, Tl, Pb, Bi) [85], and Au–In heterodimers [86]. The highest predicted EFs ranged from 10^8^–10^9^ for Al NP dimers through optimizing the size, interparticle distance, and laser excitation energy, with the best EF of 10^9^ predicted for 30 nm (radius) spherical Al NP dimers with a 1 nm interparticle spacing at λ_ex_ = 215 nm using the finite element method [76]. To demonstrate the utility of predicted EF values, we compare the predicted EF of an Al nanohole array with the experimentally measured EF from a similar Al nanohole array. The predicted Al nanohole array investigated the effect of periodicity and oxide layer thickness on the LSPR and the SERS enhancement while fixing the hole diameter at 150 nm and the hole depth at 90 nm using the FDTD method [80]. General trends showed a redshift of the LSPR as the periodicity increased and only a slight red shift of the LSPR with oxide thickness, allowing researchers to tune the LSPR for common UV lasers. The optimal periodicity was determined to be 256 nm, with a predicted EF of 4 × 10^5^ at λ_ex_ = 211 nm and up to 10^6^ at the hot spots along the edges of the holes. The effect of a 2 nm oxide layer was predicted to reduce the EF by one order of magnitude. This report can be compared with the Al nanohole array [53] highlighted in Section 3.2.1, reporting experimental EF values up to 10^6^ for a 100 nm hole diameter with a 200 nm periodicity with resonant analytes at λ_ex_ = 266 nm. When taking into account the effects of the oxide layer (decrease) and the analyte resonance (increase), the EF between the experimental (SERRS EF ~10^6^) and predicted (SERS EF ~10^4^–10^5^) Al nanohole arrays are comparable, demonstrating the utility of UV-SERS substrate performance predictions.

#### 3.2.4. Challenges and Opportunities for UV-SERS Substrates

Determining the best-performing UV-SERS substrate is challenging, as there are obstacles to calculating accurate EFs, comparing resonant and nonresonant EFs, and using non-standardized instrument conditions for estimating LODs. The goal of our analysis was to summarize the performance of the reported UV-SERS substrates (Figure 5). This illuminates trends in substrate morphology and combinations of elements that have been successful thus far (Figure 3) and provides an overview of the potential challenges and opportunities for UV-SERS substrates. One of the main challenges is the accessibility of reproducible substrates, with most commercially available substrates only in the visible range. Another challenge is the inferior performance of the UV-SERS substrates in terms of maximal enhancement factors when compared to their visible counterparts, with EFs up to ~10^7^ in the UV and EFs up to ~10^15^ (single molecule) in the visible [32,87]. While there are challenges, there are many opportunities to explore substrate morphology and composition combinations to increase substrate performance and utilize nanofabrication methods to increase the accessibility of substrates. As noted in Section 3.2.3, employing numerical simulations of LSPR and EFs for novel substrate composition and morphology combinations can provide a high-throughput method for the design optimization of novel UV-SERS substrates. To increase the accessibility of current and new substrates, we can adapt existing facile, large-scale, and cost-effective methods for UV-SERS substrate fabrication. Some of the existing nanofabrication methods used for UV-SERS substrates include:Anodic aluminum oxidation template with electron beam evaporation [55];Deposition with annealing [56];Molecular-beam epitaxial growth followed by focused ion beam or electron beam irradiation [54];Laser interference lithography [57];Ion milling [60];Nanoimprinting [60];Electron beam evaporation [60].

## 4. Future Directions for the Field of UV-SERS

Detecting environmentally or militarily relevant analytes, including water and soil contaminants, chemical warfare agents, and explosive compounds, has been explored with visible and UV-SERS [45,88,89,90]. Compared to visible wavelengths, UV excitation affords increased Raman scattering intensities, mitigation of PL interference, and higher spatial resolution (see Section 2.1). Additionally, the UV excitation can be resonant with an electronic transition of the analyte, leading to resonance enhancement (UV-SERRS) and the acquisition of PL spectra. While visible SERS has the advantage of a big head, starting with several decades of research and development, advances in UV-SERS are not far behind. The rise of machine learning and its application to SERS [91] will enable faster development of UV-SERS applications. Machine learning algorithms will be especially useful for distinguishing Raman peaks from multiple analytes, which is essential for analyzing complex biological environments and field testing real-world samples. While visible SERS has increasingly been employed for field measurements using commercially available handheld Raman spectrometers, the high cost of visible SERS substrates makes field testing or frequent monitoring prohibitive. As discussed in Section 3.2.1, Al is the most popular UV-SERS substrate, affording reasonably high EFs and low LODs. Since Al is more earth-abundant compared to Au or Ag, substrates based on nanostructured Al have the potential to be more cost-effective compared to visible SERS substrates. The cost can especially be lowered by scaling up Al nanocrystal synthesis via flow chemistry, followed by printing onto flexible substrates to make UV-SERS swabs. Furthermore, the development of UV spectrometers (e.g., instruments manufactured by Photon Systems and Wasatch Photonics) has progressed to the point where they are now available with smaller footprints comparable to visible Raman spectrometers and at similar price points.

Another important technology area that can be positively impacted by UV-SERS is in situ and operando experimentation. Developing future catalysis, materials, and biosensing technologies requires a fundamental understanding of the underlying phenomena to effectively engineer these technologies. In situ Raman characterization of synthetic reactants and products can be exploited to monitor reaction progress and kinetics or gain insights into the reaction mechanisms. In situ UV-SERS could provide unique insights into UV-activated photocatalytic reactions at low concentrations, with the possibility of operando conditions utilizing high temperature and pressure cells. Decoupling the Raman and PL energies afforded by UV excitations provides a unique opportunity to monitor both Raman and PL signatures under reaction conditions. Developing novel materials requires understanding the structure–property relationships through characterization and performance testing. Potentially, UV-SERS could contribute to understanding these structure–property relationships through the characterization of the material composition (including surface species and oxidation), thickness effects (bulk vs. thin film), and the characterization of photophysical properties through PL measurements. Moreover, the high-energy UV excitation, combined with increased absorption in the UV, can be beneficial for engineering defects through laser irradiation. This is especially beneficial in our quest to develop new materials for extreme environments through, for example, UV-induced crosslinking of polymers while simultaneously using UV-SERS to measure and quantify the changes to the materials. In the field of LSPR biosensing, methods including refractive index biosensing, MEF, and SERS provide highly sensitive, label-free, and real-time detection of biomolecules [6]. UV-SERS substrates, especially Al substrates, are potentially useful for all three methods of LSPR biosensing while taking advantage of the resonance effects of many biomolecules in the UV region. The selective sensitivity of UV-SERS for biomolecules could be useful for discriminating between elements in a heterogeneous cell mixture, especially if coupled with machine learning algorithms to aid the data analysis.

The advantages of using UV excitation lead to unique potential sensing applications relevant to the U.S. Department of the Air Force and our society. While there are many challenges, we believe the field of UV-SERS will continue to prosper as the scientific community continues to explore opportunities for growth and potential applications.

## Figures and Tables

**Figure 1 nanomaterials-13-02177-f001:**
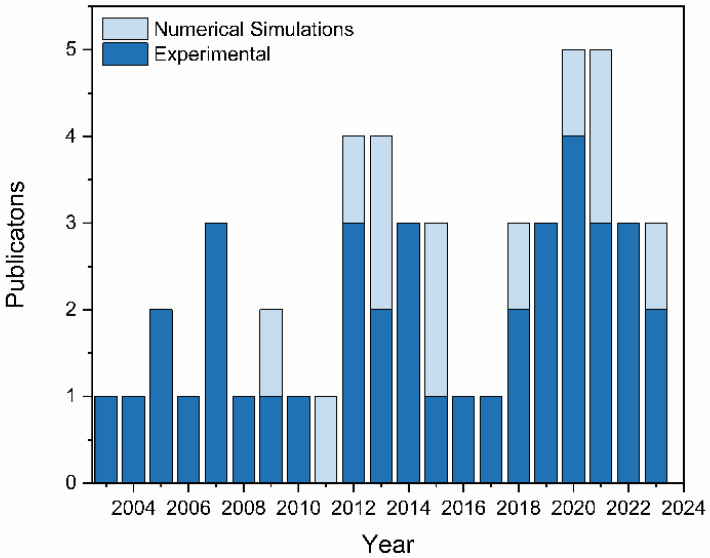
Number of publications per year separated into experimental (dark blue) and numerical simulations (light blue) reporting on ultraviolet surface-enhanced Raman scattering (UV-SERS) substrates. Total publications: 51. The following search terms were used in Google Scholar, Web of Science, and Science Direct databases: “ultraviolet surface-enhanced Raman scattering,” “UV SERS,” “deep ultraviolet surface-enhanced Raman scattering,” “DUV SERS,” “ultraviolet surface enhanced resonance Raman scattering,” “UV SERRS,” “deep ultraviolet surface enhanced resonance Raman scattering,” OR “DUV SERRS.” The most recent search was on 8 June 2023.

**Figure 2 nanomaterials-13-02177-f002:**
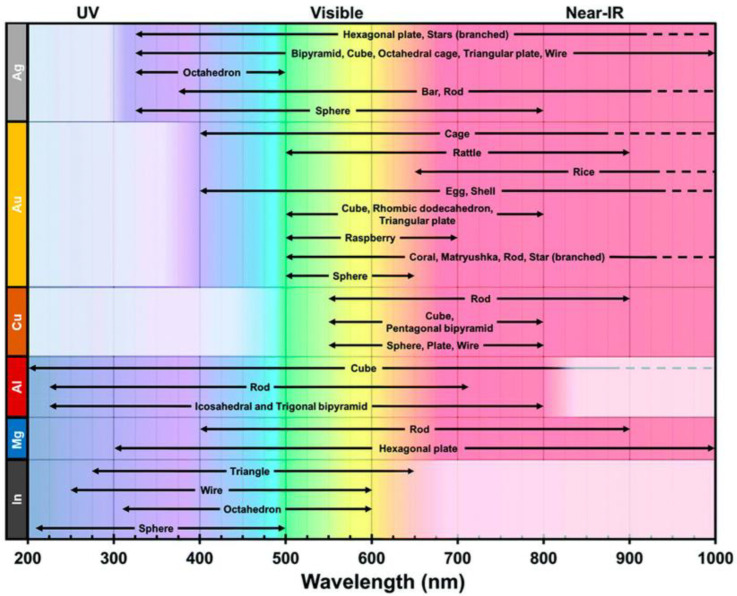
Summary of typical LSPRs reported in the literature includes several elemental composition and morphology combinations that span from the UV to the NIR. Reprinted with permission from Ref. [34]. Copyright 2022, American Chemical Society. The original figure was adapted from Ref. [15] with permission from the Royal Society of Chemistry.

**Figure 3 nanomaterials-13-02177-f003:**
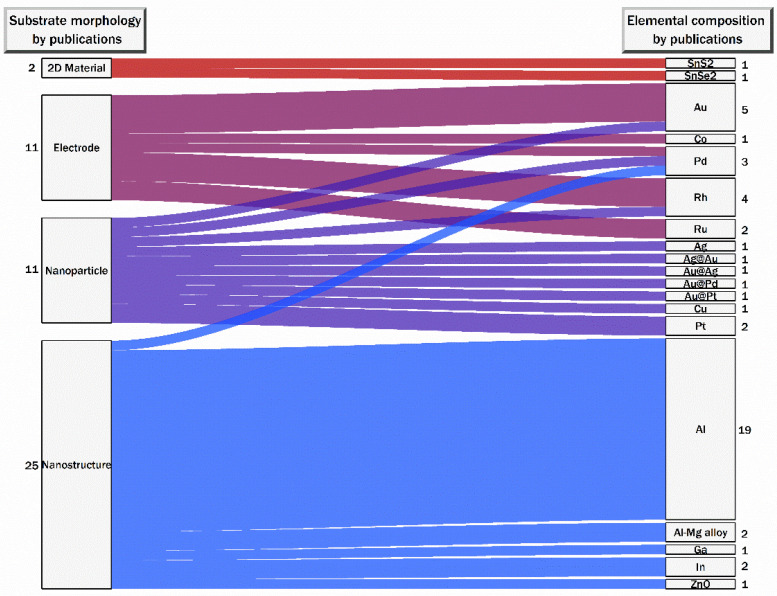
Analysis of reported UV-SERS substrates demonstrating the substrate morphology (left column) with the corresponding number of publications and the elemental composition (right column) with the corresponding number of publications. The thickness of the colored links through the middle illustrates the frequency of a particular substrate morphology and elemental composition combination, such as Al nanostructures, with the most reports analyzed (19 publications).

**Figure 4 nanomaterials-13-02177-f004:**
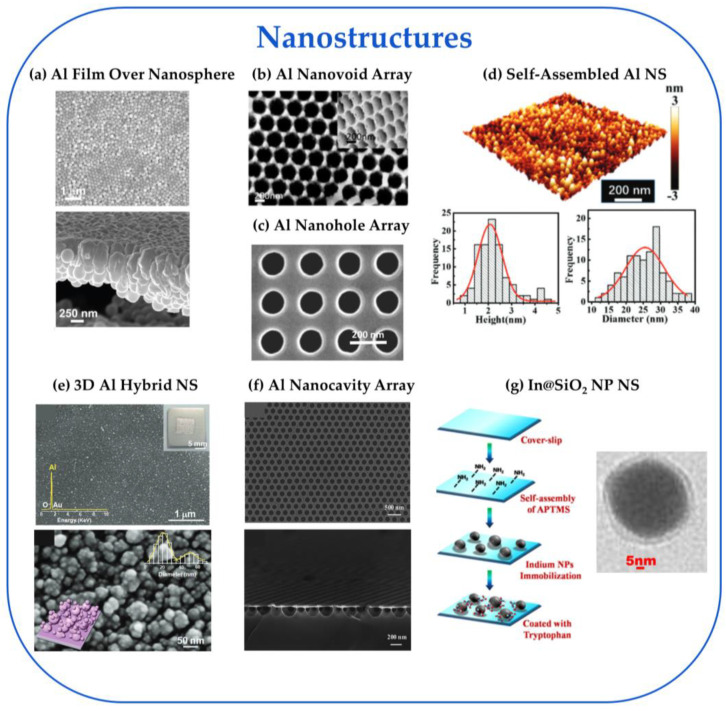
A visual representation of the variety of UV-SERS substrates in the nanostructure morphology category: (**a**) SEM image of Al film over nanosphere (FON) taken at 15,000× (top) and 50,000× (bottom). The Al film was 200 nm thick on 210 nm carboxylated latex/polystyrene spheres. Reprinted with permission from Ref. [49]. Copyright 2016 American Chemical Society; (**b**) SEM image of the top view of Al nanovoid array (NVA) with 45° view in inset constructed with a 400 nm thick Al film with 200 nm void diameter. Reprinted with permission from Ref. [62]. Copyright 2013 American Chemical Society; (**c**) SEM image of an Al nanohole array (NHA) with 100 nm hole diameter, a 200 nm periodicity, and 150 nm thick Al film. Reprinted with permission from Ref. [53]. Copyright 2021 American Chemical Society; (**d**) AFM image of self-assembled NS with stalagmite-shaped Al NPs that were annealed at 200 °C for 900 s (top) and the resulting Al NP height and diameter distribution histograms (bottom). Reproduced from Ref. [56] with permission from the Royal Chemical Society; (**e**) Low-magnification SEM image with EDS spectrum for the 3D Al hybrid NS (top) and high-magnification SEM image of the 3D Al hybrid nanostructure with a schematic illustration of the structure in the lower left corner (bottom). The inset shows the size histogram of the Al NPs with the corresponding Gaussian fits, where the larger Al NPs are 50 nm and the smaller Al NPs are 20 nm. Reproduced with permission from Ref. [46]. © 2017 WILEY-VCH Verlag GmbH & Co. KGaA, Weinheim; (**f**) Top (top) and cross-section (bottom) view SEM images of Al nanocavity array (NCA) with a 350 nm average diameter, a 500 nm periodicity, and 50 nm thin Al film. Reprinted with permission from Ref. [59]. Copyright 2021 American Chemical Society. (**g**) Schematic for the self-assembly on spherical In NPs coated SiO_2_ (left). Magnified TEM image of 40 nm In NP with 2 nm SiO_2_ coating (right). Reproduced with permission from Ref. [65]. © 2019 Elsevier B.V. All rights reserved.

**Figure 5 nanomaterials-13-02177-f005:**
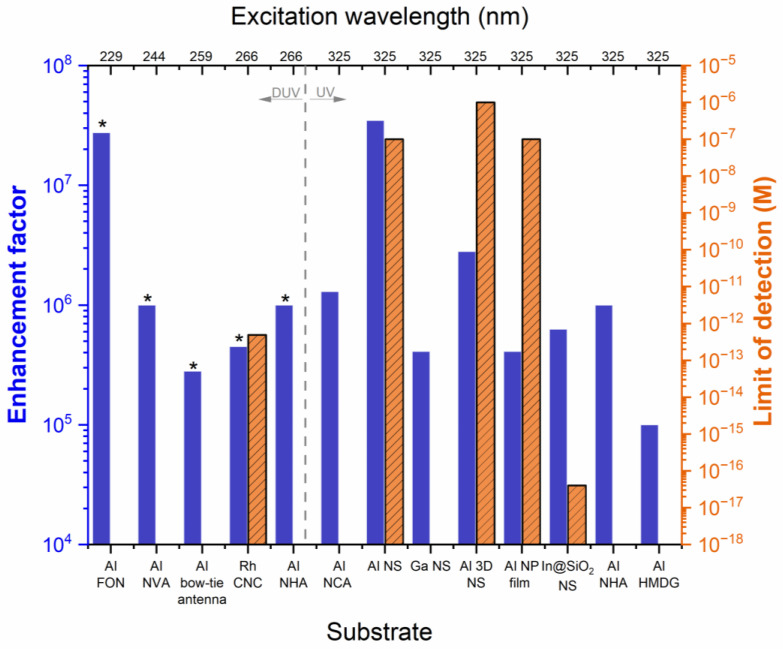
Analysis of the reported UV-SERS substrate performance. The left (blue) y-axis plots the reported enhancement factor, and the right (orange-striped) y-axis plots the reported limit of detection (M). The bottom x-axis gives a brief description of the substrate, with the following abbreviations: film over nanosphere (FON); nanovoid array (NVA); concave nanocube (CNC); nanohole array (NHA); nanostructure (NS); nanoparticle (NP); hybrid metal-dielectric grating (HMDG). The top x-axis provides the laser excitation wavelength, with the gray dashed line dividing the DUV and UV regions. An asterisk (*) denotes if a resonant analyte was used in the reported performance metrics. The citations for the DUV substrates from left to right: Al FON [49], Al NVA [62], Al bow-tie antenna [61], Rh CNC [45], Al NHA [53]. The citations for the UV substrates from left to right: Al NCA [59], Al NS [56], Ga NS [64], Al 3D NS [46], Al NP film [50], In@SiO_2_ [65], Al NHA [47], Al HMDG [57].

**Figure 6 nanomaterials-13-02177-f006:**
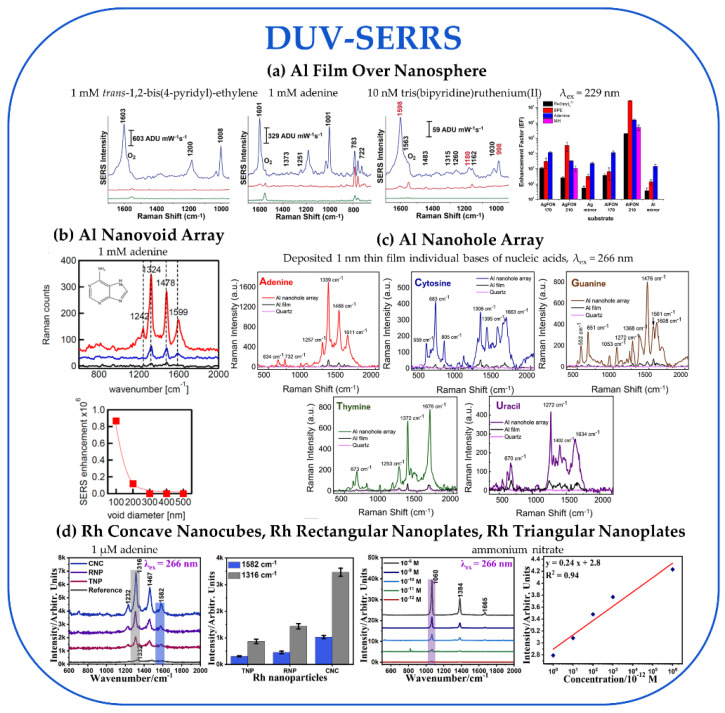
DUV-SERRS experimental data are discussed in Section 3.2.1. All spectra have the laser excitation wavelength and the analyte(s) concentration noted in the figure. (**a**) Raman spectra for the Al film over nanosphere (FON) substrate with three analytes, where the blue spectra are for the 210 nm particles, the red spectra are for the 170 nm particles, and the green spectra are on quartz slides. The rightmost panel is a summary of the reported EFs for each substrate/analyte combination studied. Reprinted with permission from Ref. [49]. Copyright 2016 American Chemical Society; (**b**) Raman spectra for adenine on the 200 nm Al nanovoid array (red), an evaporated Al surface (blue), and the adenine solution without a plasmonic substrate (black) are shown in the top panel. The bottom panel summarizes the reported EFs as a function of nanovoid diameter. Reprinted with permission from Ref. [62]. Copyright 2013 American Chemical Society; (**c**) Raman spectra for five nucleic acid bases on an Al nanohole array with a 100 nm diameter and a 200 nm periodicity compared with the spectra with the respective nucleic acid base on an Al film substrate and on a quartz substrate. Reprinted with permission from Ref. [53]. Copyright 2021 American Chemical Society; (**d**) Leftmost panel shows the Raman spectra of adenine with the Rh concave nanocubes (CNCs), Rh rectangular nanoplates (RNPs), and Rh triangular nanoplates (TNPs). The next panel compares the intensities of the two most prominent vibrations across the three substrates. The third panel displays the Raman spectra for ammonium nitrate at various concentrations with the Rh CNC. The last panel shows a linear relationship between the concentration of ammonium nitrate with Rh CNC and the Raman intensity used to determine the LOD. Reprinted with permission from Ref. [45]. © 2022 John Wiley & Sons Ltd.

**Figure 7 nanomaterials-13-02177-f007:**
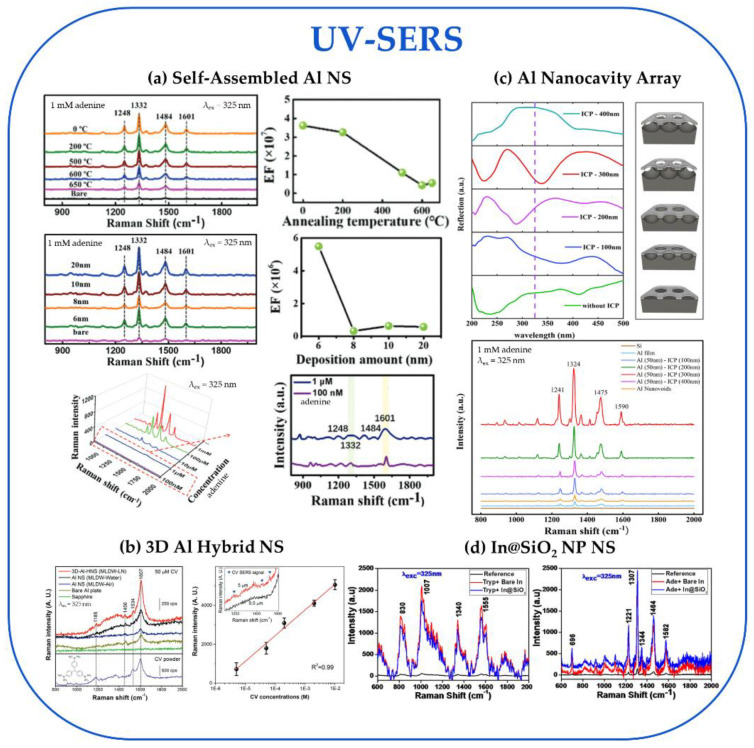
UV-SERS experimental data are discussed in Section 3.2.2. All spectra have the laser excitation wavelength and the analyte(s) concentration noted in the figure. (**a**) The first row shows the Raman spectra of adenine on the self-assembled Al NS as a function of annealing temperature and a summary of the reported EF values. The second row displays the Raman spectra of adenine on the self-assembled Al NS as a function of Al deposition thickness and a summary of the reported EF values. The third row shows the Raman spectra for adenine at various concentrations to determine the LOD. Reproduced from Ref. [56] with permission from the Royal Chemical Society. (**b**) Raman spectra of crystal violet on the 3D Al hybrid NS substrate, control Al NS substrates, a bare Al plate, sapphire, and the bulk powder for comparison (left). The right panel shows the relationship between the concentration of crystal violet on 3D Al hybrid NS (MILDW-LN) and the Raman intensity that was used to determine the LOD. Reproduced with permission from Ref. [46]. © 2017 WILEY-VCH Verlag GmbH & Co. KGaA, Weinheim. (**c**) Reflectance spectra (top) of the Al nanocavity arrays with different inductively coupled plasma (ICP) etching depths and a schematic of each substrate. The Raman spectra (bottom) for adenine on the Al nanocavity arrays with a 50 nm Al film and different ICP etching depths were compared to an Al nanovoid array (no etching), an Al film, and Si as controls. Reprinted with permission from Ref. [59]. Copyright 2021 American Chemical Society. (**d**) Raman spectra of tryptophan (left) and adenine (right) at 40 nm In NPs coated with 2 nm of SiO_2_ and compared to bare In and a reference. Reproduced with permission from Ref. [65]. © 2019 Elsevier B.V. All rights reserved.

## Data Availability

No new data were created or analyzed in this study. Data sharing is not applicable to this article.

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
