# Peer review of "Beyond the Visible: A Review of Ultraviolet Surface-Enhanced Raman Scattering Substrate Compositions, Morphologies, and Performance"

_nanomaterials, 2023, doi:10.3390/nano13152177_

Round 1

Reviewer 1 Report

This paper is a stimulating review on the advantages of surface-enhanced Raman scattering under ultraviolet excitations (UV-SERS). It is a matter of fact that this technique is insufficiently known an used by optical spectroscopists. As a result, the majority of the groups interested by SERS use visible and NIR (near-infrared) excitations using adequate substrates for them. The paper explains clearly with the help of graphics which substrates should be used in order to extend their future explorations toward UV-SERS technique. 

Author Response

We would like to thank the reviewer for their comments. There were no recommended revisions from the review report. 

Reviewer 2 Report

In this review, the authors evaluated the reported UV-SERS substrates in terms of their elemental composition, substrate morphology, and performance in the past 20 years. There are some specific comments for this manuscript:

1. For the abstract, the authors stated the main challenge of the UV-SERS substrate that resulted in its slower development compared to its visible and NIR counterparts. The authors are suggested to claim the significance of exploring UV-SERS substrate and its applications.

2. For Figure 1 caption, “DUV SERRS” should be placed behind the "deep ultraviolet surface enhanced resonance Raman scattering".

3. The font in Figure 4 caption needs to be consistent.

4. Recent related papers have reported magnetic-response SERS nanomaterials (Theranostics 12 (13), 5914) and multifunctional SERS-active nanoplatform for drug delivery and cancer therapy (Small, 2023, 2206762). The authors are advised to cite these related papers accordingly.

Author Response

We would like to thank the reviewer for their comments and suggestions. Below are point-by-point responses in green. 

  1. For the abstract, the authors stated the main challenge of the UV-SERS substrate that resulted in its slower development compared to its visible and NIR counterparts. The authors are suggested to claim the significance of exploring UV-SERS substrate and its applications.
    • We added 4 sentences to the abstract (highlighted in yellow) to address the advantages of UV excitation, top performing substrate, and explicitly state the applications discussed. 
  2. For Figure 1 caption, “DUV SERRS” should be placed behind the"deep ultraviolet surface enhanced resonance Raman scattering". 
    • Changed and highlighted figure caption in yellow
  3. The font in Figure 4 caption needs to be consistent.
    • Good catch. We fixed the in-text citation font size and found the same error in Figures 2, 6, & 7. We highlighted the figure captions in yellow. 
  4. Recent related papers have reported magnetic-response SERS nanomaterials (Theranostics 12 (13), 5914) and multifunctional SERS-active nanoplatform for drug delivery and cancer therapy (Small, 2023, 2206762). The authors are advised to cite these related papers accordingly.
    • We read through the very interesting SERS articles cited by the reviewers. We did not include these articles because they used NIR excitation (785 and 808 nm), which is outside of the scope of this review. Upon further review, we still feel these articles are outside the scope of the review on UV-SERS and decided not to include them.